# The Lived Experiences of Women without COVID-19 in Breastfeeding Their Infants during the Pandemic: A Descriptive Phenomenological Study

**DOI:** 10.3390/ijerph19159511

**Published:** 2022-08-02

**Authors:** Ka-Huen Yip, Yuk-Chiu Yip, Wai-King Tsui

**Affiliations:** Caritas Institute of Higher Education, School of Health Sciences, 2 Chui Ling Lane, Tseung Kwan O, New Territories, Hong Kong SAR, China; jeffreyycyip@gmail.com (Y.-C.Y.); ztsui@cihe.edu.hk (W.-K.T.)

**Keywords:** burnout, COVID-19 pandemic, postnatal care, psychological distress, psychosocial support

## Abstract

The coronavirus disease 2019 (COVID-19) outbreak in 2020 has led to several changes and disturbances in the daily lives of the general public. Particularly for new (first-time) mothers, there has been a significant impact on the practices of raising and feeding their babies. Social distancing measures everywhere have made mothers hesitant to breastfeed their babies anywhere else but at home. Combined with the fear of being infected with COVID-19, the present situation has created unprecedented barriers for breastfeeding mothers to accessing various types of support: emotional, instrumental, informational, and appraisal. There has been no research on the influence of the pandemic on social support regarding breastfeeding in Hong Kong. This study aimed to explore the social support and impact of COVID-19 on mothers breastfeeding their babies. Semi-structured interviews were conducted with 20 currently breastfeeding women in Hong Kong. Colaizzi’s seven-step method was used for data analysis. Two key themes emerged from the interview data: (1) positive influences on breastfeeding support during COVID-19 and (2) negative influences on breastfeeding support during COVID-19. Our findings may help mothers prepare to breastfeed their babies in places other than their homes.

## 1. Introduction

The coronavirus [SARS-CoV-2] disease or COVID-19 rapidly spread, infecting over five hundred million people worldwide and causing over six million deaths [1]. People with COVID-19, as an acute respiratory, disease may have varied symptom severity [2,3]. The World Health Organization declared COVID-19 a public health emergency of international concern in January 2020 [4,5]. According to studies, the low mortality rates of people under 65 years, and those with no underlying health conditions, highlight that the death rates are higher among individuals with clinical vulnerabilities [6,7,8]. There is a higher predisposition to contract COVID-19 during the third trimester of pregnancy, and among early postpartum women [9,10,11]. All of these women are advised to seek urgent medical advice and care and stay at home to protect themselves and their babies, to prevent preterm birth and pregnancy loss during the COVID-19 pandemic [12,13]. However, this vulnerable group of women suffer a greater impact of COVID-19 during pregnancy compared to other groups, especially during the final trimester [9,14,15,16]. The main impact concerns the awareness of social support and the impact of COVID-19 on mothers breastfeeding their babies, and the need to seek more understanding about how to provide better resources and emotional support for those breastfeeding women, thus decreasing their stress and improving relationships.

### COVID-19, Social Distancing Restrictions, and Postnatal Vulnerabilities

Although breastfeeding initiation rates are high in Hong Kong, where over 87% of mothers initiate breastfeeding, only half of these mothers exclusively breastfeed their babies [17]. Moreover, this 2019 survey reported that Hong Kong demonstrates a big drop in the number of mothers who practice exclusive breastfeeding in the first two months following birth, while the proportion of infants who are exclusively breastfed in the first 4–6 months is low [17]. In 2020, the Hong Kong Government increasing statutory maternity leave to 14 weeks, and forbidding discrimination on the ground of breastfeeding, was a new contribution to breastfeeding progress [18] and increased the number of Baby-Friendly Hospitals to deliver the continuum of care that supports mothers to feed their babies optimally [19]. Breastfeeding in public is still considered taboo; 40% of breastfeeding mothers who fed their babies in public have encountered uncomfortable conditions [20]. However, breastfeeding in Hong Kong is hard; most of the women who discontinue breastfeeding do not terminate voluntarily but due to breastfeeding issues, including the lack of preparation for breastfeeding, in-hospital support services, post-natal support, and public areas for breastfeeding [21,22].

With the higher transmissibility and risk of infection of the omicron strain, on 3 March 2022 [23] the Hong Kong Government strongly urged its citizens to abide by social distancing measures, to avoid going out, and to abstain from participating in unnecessary or crowded activities and gatherings, such as religious or family activities. A similar stay-at-home order was implemented by the United Kingdom Government [24]. This would prevent the spread of COVID-19 and limit COVID-19 deaths in the community.

Studies have reported that pregnant women can easily be infected during epidemics [25,26,27]. During the severe acute respiratory syndrome (SARS) outbreak in Hong Kong [28], women described their psychological distress, such as frustration, anxiety, sleeping disturbance, and interference in their daily lives. The postnatal period is an important transition period for women and can alter many aspects of life, including the role and identification of the infant’s mother, motherhood [29], lifestyle [30], mother-infant bonding, confidence, and satisfaction [31]. Postnatal mothers may, in part, face an increased risk of experiencing mental distress during the COVID-19 pandemic [32]. Hence, transitioning into motherhood during COVID-19 presents unique stressors, which may worsen an already vulnerable period in a woman’s life [33]. The findings of a qualitative study that explored the perinatal experiences of mothers during COVID-19, highlighted that virtual consultations from healthcare professionals were viewed as impersonal and caused women to feel too embarrassed to talk about their mental health concerns [34].

A web-based survey investigated the psychological experiences of women during COVID-19 [35]. A rapid literature review on the impact of COVID-19 on maternal mental health and perinatal mental health services, between October 2019 and September 2020, indicated depression and anxiety levels among mothers before October 2019 as approximately 11% and 18%, respectively [32]. A percentage of anxious mothers attained above the relevant cut-off for September 2020 indicated levels of depression (43%) and anxiety (61%) in clinical settings. A large percentage of mothers felt worry and panic because they may not have adequate support to fulfill their needs. There are only a few qualitative studies that have assessed women’s psychological experiences during the COVID-19 pandemic [36,37]. Most literature has focused exclusively on the higher transmissibility and risk of infection of the omicron strain in Hong Kong [35,36,37]. Using qualitative research methods can provide a richer and more in-depth insight into which elements of social distancing restrictions have generated an impact on mothers’ emotional health. The experiences of women breastfeeding their babies, while not being infected with COVID-19, have not been investigated. Therefore, this study aimed to explore the in-depth experiences of this vulnerable group of women during the COVID-19 pandemic in Hong Kong, with regards to their psychological state, perspectives, feeding methods, personal life, the experience of home quarantine, and the impact of COVID-19 on their mental health.

## 2. Materials and Methods

### 2.1. Ethical Review

The participants were informed of the details of the study and informed consent was obtained from all of them. Information about the study was provided to the participants before conducting interviews, including its background, purpose, and procedure. Participation was voluntary, and all participants were informed that they could withdraw from the study without any consequences. The confidentiality and anonymity of the participants were strictly ensured through encryption. Only the researchers had access to the study data. Ethical approval was obtained from the Caritas Institute of Higher Education, Research and Ethics Committee (HRE210136).

### 2.2. Design

A qualitative research design was used and data were collected through individual, semi-structured interviews, on the women’s psychological experiences with breastfeeding their babies during COVID-19. The data were analyzed based on the phenomenological methodology suggested by Colaizzi [38]. With this method, the researchers attempted to understand the participants’ subjective feelings and experiences by having them mentally return to the situation itself.

### 2.3. Participants

The participants in this study were 20 women breastfeeding their babies in Hong Kong. Purposive sampling was used, and to recruit participants our research team approached peer support groups from online media platforms such as Facebook, and groups related to mothers in Hong Kong, such as “Hong Kong Moms,” “Little Steps,” and “Mother Kingdom.” The interviews were conducted by telephone or video calls (e.g., Zoom). The research team members then individually followed up with the participants who showed interest in participating in this study, who obtained our contact information (e-mail and/or phone) through these online platforms.

All participants met the following inclusion criteria: (1) women having experiences in providing their child with breast milk in any form (such as direct breastfeeding or using a bottle to feed their babies with their own milk) at least once during the COVID-19 pandemic, (2) women having lived in Hong Kong during the 12-month period as aforementioned, and (3) women whose babies were within an age range of 0 to 12 months.

Individual interviews continued until data saturation was reached at the 17th interview, after which no new information arose in the next three individual interviews. Table 1 shows the participants’ characteristics. We enrolled 20 women between the ages of 22 and 46, with an average age of 32 years, who breastfed their babies during the COVID-19 pandemic. Among the participants, two women completed secondary education, nine possessed a bachelor’s degree, three possessed a master’s degree, and six graduated with a higher diploma. Regarding the breastfeeding period, 12 participants were first-time mothers using different feeding methods, seven participants had two children, and one participant had three children.

### 2.4. Data Collection

From December 2021 to February 2022, all interviews were conducted in Chinese by the researcher (W.K.) at the participant’s preferred time schedule in a private password-protected Zoom (Nasdaq, San Jose, CA, USA) meeting room. Each interview took about 60–130 min, and the same interview guide was always used to reduce variation in the data collection process. In addition, field notes were taken during the interviews to facilitate the collection of contextual information for data analysis. The interview guide questions are presented in Table 2. An academic qualitative scholar and a psychological consulting specialist were employed to form an expert panel to review and affirm the validity of the interview questions. All participants’ interviews were audio-recorded, with their written consent. If the participant exhibited emotional distress during the interview, adequate psychological support was provided to prevent further psychological harm.

### 2.5. Data Analysis and Trustworthiness

In addition, data analysis was conducted immediately after data collection using NVivo Version 12, QSR International [39]. Two research team members (K.H.Y. and Y.C.Y.) transcribed the interview tapes verbatim and, subsequently, translated all transcripts into English. Back translation was undertaken by another team member (W.K.T.) and a professional translator to ensure semantic equivalence. The data and field notes of all interviews were organized systematically according to Colaizzi’s phenomenological analysis method [38]. This method follows a seven-step approach to expose emergent themes, namely (i) to familiarize the researchers with the collected data, (ii) to identify the significant statements, (iii) to formulate the meanings and the use of reflection by the researchers, (iv) to cluster and define themes, (v) to formulate an exhaustive description, (vi) to create a fundamental composition of the phenomenon, and (vii) to verify the exhaustive description and fundamental composition [38].

In particular, all recorded interview data were transcribed verbatim and analyzed throughout the data analysis process [38]. All the interview data included meaningful statements and gave rise to different themes, which were independently reviewed by two research team members (K.H.Y. and Y.C.Y.). They looked at all the analyzed data and compared the subthemes, identifying the differences and similarities between them. The authors of this study (K.H.Y., Y.C.Y., and W.K.T.) discussed the results of their analyses to reach a consensus. The study referenced the stringent criteria established by Lincoln and Guba to ensure a high level of rigor in this study [40,41,42]. It demonstrates how each criterion regarding credibility, dependability, confirmability, and transferability was achieved through member checking.

Given credibility in creating confidence that the results are accurate, true, credible, reliable, and believable in: (1) prolonging and diversifying engagement with each composition, (2) having interviews with comprehensive process and techniques, (3) building investigators’ authority, (4) gathering all referential adequacy materials, and (5) having peer debriefing sessions regularly.

Engagement through participant observation in the field was infeasible at the time of this research. There was a strict “multi-household gatherings at private premises” policy (as imposed by the Government of Hong Kong Special Administrative Region) that prohibited more than two households and groups gathering in the community. This policy was to prevent the spread of the virus to the community. However, several participants from different living and working venues were approached through peer support groups that had regular communications about the research with the authors’ affiliated institution. After being granted ethical approval of this study, the interview guide was assessed at two induction meetings. Two pilot interviews via Zoom were performed, and these two interviews’ data were also integrated in the final data analysis. All research team members had the required data management skills, knowledge, and practical experience and involvement of more than 4 years in qualitative research to conduct their roles.

Furthermore, field notes were applied as a tool to analyze the transcripts and utilized to aid the documentation of the contextual information mentioned by the participants for accurate data analysis. Moreover, the research team members were scheduled for debriefing sessions held regularly at 2-week intervals with the Fellows from the Hong Kong Academy of Nursing, to guarantee that there were no taken-for-granted biases, attitudes, perspectives, or assumptions on the part of researchers.

Enhancing dependability to guarantee the results of this qualitative inquiry would be duplicable if the inquiry existed within the same cohort: (1) to apply rich description of study methods, (2) to set up an audit trail, (3) to progress in a series of distinct stages in the replication of data.

The research team tried to present the study methods in detail and clearly in the research papers; they set up a detailed track-record of the data-collection process by all research team members. Additionally, to ensure the validity of the accounts from the participants, member-checking was conducted to enhance the clarity of the meanings derived from the participants. Research team members read the transcripts to the participants via telephone to clarify the interpretative meanings. It provided participants the opportunity to recognize and accept their narrative contributions, once they have been put into the research report in rough and final draft forms. All research team members appraised coding accuracy and inter-coder reliability throughout the data-analysis process.

Confirmability was maintained by empowering confidence that the results would be confirmed/corroborated by other researchers; it had self-reflection and reflexivity. This approach evaluates reflexive journals and weekly investigator meetings were applied.

Moreover, transferability was enhanced by developing the degree to which the results can be summarized/transferred to further contexts/settings; it was achieved through data saturation and possessing thick description. This approach comprised two main issues: (1) data saturation was satisfied when no new themes appeared from the participants; all research team members gained a consensus on the attainment of data saturation; (2) lengthy description was presented in the quotes of the participants, such that the meanings of the narrative statements from the participants could be translated and interpreted in context.

## 3. Results

Using phenomenological methods, we explored the psychological experiences of women breastfeeding their babies during the COVID-19 pandemic. Two themes were identified from our observations (summarized in Table 3).

### 3.1. Theme 1. Positive Influences on Breastfeeding Support during COVID-19

Participants were asked a series of questions about their daily activities and how they cared for their child while breastfeeding during the COVID pandemic, including how various personal protective measures and restricted movement to curb the spread of the virus had affected their breastfeeding and motherhood experience.

#### 3.1.1. Mothers’ Desire for Support

Participants described both positive and negative breastfeeding experiences due to the COVID-19 pandemic. They did not generally tend to need support for steps toward a good breastfeeding latch. However, regarding empathic and understanding support, they had established friendships and peer support groups to share their questions or concerns related to breastfeeding and caring for the baby, which, in turn, aided their ability to be open and frank. Some participants tried to search for more information, including special Chinese traditions recommended for postpartum dietary practices during this period.

“As a breastfeeding mother, I recognize that I need a lot of support in the early stages of feeding. Thank goodness my son has grown up and I do not need any support during COVID-19.”

“In that time, I felt alone. She (breastfeeding mothers from peer support groups via online platform) felt like someone who is close to me and gives me all experience that I need. It was almost like we were friends. I did not even know her face-to-face. However, I really valued what she said; I would call it a friendship, I would think.”

“If I had a young child and was without any support, I would think hard to decide if I should continue breastfeeding. I am lucky to have joined the ‘Hong Kong Moms,’ which is an online social platform group for breastfeeding mothers. I came across other breastfeeding mothers in this baby group in this online social platform during COVID-19. I mean it is different because I know I am talking to someone whom I only know by their nickname, someone who has experienced it on some level, which I think is very important, and is helpful with different kinds of advice and neutral opinions, such as how to deal with their own disasters. Besides, being anonymous, I think, was also important. It was nice to have that support … I have searched some interesting new ways of ginger vinegar soup and pig’s trotter (delivering service), online or by phone, which I shall continue to establish and use in the future. I do not need to worry about it all, and take my time for rest.”

“In the mothers’ group, we would chat mostly about breastfeeding, although we also chatted about other topics. They would inquire how my baby was sleeping and provided help that way; they were reliable. The other mothers in my mothers’ group were coping with their own catastrophes. It was wonderful to have that support.”

During the analysis of other interviewees’ responses, the findings relating to many subthemes expanded. It was interesting to note how a similar issue that took place for various interviewees could often be interpreted as either positive or negative, depending on the individual and their perception of the circumstances.

Some participants described that they had more privilege in their living environments without financial burdens. They enjoyed the breastfeeding journey with a better living environment and family support, which impacted the health outcomes of breastfeeding and baby care. With a shift in the mindset of men toward family care, some participants reported that their partners were willing to stay at home longer to take care of their newborn babies. Some fathers were granted paternity leave and spent extra time to support their spouses in breastfeeding and maternal recovery, both physically and emotionally. Some participants’ partners were working from home and could spend more time being present than if they were out at work all day. This shared and collaborative care increased the bonds between partners and their babies, and strengthened the relationship between the parent and infant:

“My husband is working from home because of COVID-19 and has taken 10 days off. He enjoys so many special moments with us every day. He can help with diaper changes and baths, take care of our baby, and let me sleep when I need it … Our family lives in a house of two floors. My husband slept in the other bedroom. He is afraid of affecting my rest and lets me have enough space to rest and exercise. Maya (Filipino domestic helper) also helped to cook lot of ginger, eggs, and chicken to increase the nutritional content in my diet. Basically, my husband and I share every little bit of happiness that is happening in our lives. The two of us have been here with each other for the baby, which means I have more time to focus on breastfeeding our baby.”

#### 3.1.2. COVID-19 Is Not All Bad for Mothers

Some participants stated that during the social distancing period, they could only allow a few visits from their friends and relatives to greet their newborn babies. They could therefore relax at home and spend more time attending to their babies, and rest instead of entertaining many friends and relatives eager to see their babies. Besides, with fewer visitors, there was less unsolicited advice, and the mothers could focus on their own way of breastfeeding their babies:

“Oh! I recall being overwhelmed by my family and friends visiting me and my son during the first to fourth weeks after giving birth to my first son. They felt that I did not know how to breastfeed my son because my son often burst into tears. When they wanted to help comfort my son, I was often unable to react at that time. As such, my son could easily become irritated most of the time, which was not good for his mental growth. At that time, I especially felt a lot of pressure from my grandma and I needed to supplement my breastfeeding with formula milk. Now that I am breastfeeding my second son during COVID-19, I do not have that stress, and this time I can really feel the enormous difference in feeding him and my mental health.”

Several participants expressed that during the period of social distancing, they could experience a slow pace of life. There was no place to go and hence, they would have all the time to focus on how and when to breastfeed their children. Some participants, especially those who were struggling with the pain of bleeding nipples, breast engorgement, stress, and anxiety, welcomed this peace of mind. They believed that before social distancing, they would be busy going out and meeting friends and family to introduce them to their babies. As they would still be experiencing the pain and pressure of visiting friends and family, they would rather stop breastfeeding.

“I find it very difficult for me as a breastfeeding mother because I have breastfeeding problems, such as bleeding nipples, tearing, and a lot of pain. Having been home for maternity leave because of COVID-19, I was able to spend time to focus on breastfeeding; my daughter is the only reason I persist in continuing breastfeeding. I am having a lot of visits from family, mom, or other friends and I need to put up with the stress that they cause to me. If it were not for the COVID-19 pandemic to reduce social gatherings, I am not sure I would be successful in breastfeeding my little daughter. I now have had six weeks of feeding!”

Coupled with the aforementioned factors, most participants were also concerned about maintaining their own privacy during breastfeeding. All participants were Chinese, and the common reason they had for stopping breastfeeding sooner than planned was the discomfort and embarrassment of feeding in front of somebody else.

Participants expressed that they felt at ease at home. On the contrary, they were embarrassed and uncomfortable breastfeeding their baby this way outside, in front of visitors, or in public. Several participants were accustomed to feeding their baby on a comfortable chair, shirtless, and through a lot of skin-to-skin contact with the baby. This allowed them to gain breastfeeding confidence to position and support their baby for a good latch, the most natural and beneficial activity for the mother and baby.

“Not having family and friends visiting me after giving birth to my baby has allowed me time to practice how to breastfeed my daughter without having to worry about people coming, and not being able to go out during the COVID-19 pandemic has given and enhanced my confidence in breastfeeding. Having said that, I am scared and do not have faith to take my daughter to shopping malls and parks and feed her in public. I think there should be suitable and appropriate facilities for privacy for breastfeeding in the malls and parks before I breastfeed my daughter in those areas.”

During the COVID-19 pandemic, participants felt that they had extra time without any pressure regarding breastfeeding. They could perform more natural feeding in responding to hunger cues and satiety from their baby, rather than following a mother-led routine. Some participants also appreciated the additional help from healthcare professionals via online communicating support systems, granted without social isolation during the COVID-19 pandemic. Participants stated that they breastfed their babies more often due to no planning in their daily activities; for instance, they did not need to plan things like going out to work on time or getting home quickly. This positively affected perceptions of milk supply by mothers and increased the early weight gain of their baby.

“I am working from home during COVID-19, so I have more time to focus on feeding my little princess as and when she wants to be fed, and not feeling rushed because I do not need to go anywhere … It was nice to have somebody who could maybe talk through text-based communication on mobile applications. I really valued that this support was able to listen and understand what I said. They can absolutely direct me to correct information without any confusion. Moreover, through interactions, I had no feelings of social isolation, and it also helped me relax.”

Some participants stated that their return to work after maternity leave collided with the worsening of the COVID-19 pandemic. Some companies allowed employees to work from home. This relieved the mothers’ worries about sending their babies to a nursery, or asking family members to take up babysitting while they were away for work. The mother could have more intimate contact with her baby and more breastfeeding; hence, the baby could have more breast milk rather than formula milk. The participants also felt relaxed under such work arrangements.

“I am going back to work next Monday after maternity leave, and I do not actually know what uncontrollable circumstances this will bring. Despite the public panic of the COVID-19 pandemic, it is unlikely I will not be at work. This means relief from anxiety, and I can continue my breastfeeding because after maternity leave, the company allowed me to work from home, so I am still at home. I do not need to ask my mother to come over to my house to take care of my daughter. At the same time, I do not need to worry about pumping breast milk as reserve for feeding my daughter.”

### 3.2. Theme 2. Negative Influences on Breastfeeding Support during COVID-19

Various participants explained the pessimistic impact on their own breastfeeding experience during the COVID-19 pandemic.

#### 3.2.1. Absence of Close and Personal Professional Coaching

During the stay-at-home requirement during the COVID-19 pandemic, most mothers with first babies came across a common problem of having less counseling support in breastfeeding. Instead of healthcare providers standing close by during breastfeeding, the participants could only describe their difficulties through phone calls and online messaging. They could not come near, observe, and note the incorrect breastfeeding steps or practices of the new mothers. This is considered a significant barrier for first time mothers to start breastfeeding. The new mothers were concerned about the lack of personal support and only having support through phone calls.

Most participants really missed having someone who could look at what was happening and help them make small changes to improve their breastfeeding. This feeling meant that the participants lacked close and professional coaching during breastfeeding. They expressed a sense of sadness, anxiety, and unhappiness at losing the idea of breastfeeding they hoped to have.

“The breastfeeding class was cancelled due to the COVID-19 pandemic. He is my first child; I have no experience at all. I feel I am not breastfeeding properly. I do not know what to do, I am depressed and retarded, and I am scared that my baby needs to lose weight. I do not stop using my mobile phone to search for a lot of information via the Internet. I do not know if it is right or wrong, and I am afraid that my baby will be infected with COVID-19. I am experiencing great levels of anxiety, stress, and uncertainty … I cannot have a get diagnosis due to the current situation. I am also experiencing a lot of pressure from the stress that my baby might have to be hospitalized because of his excessive weight. This has been a very difficult process, and I have been in great pain and confusion. Three days since we have returned home, we have received no further support. I was crying on the phone when asking for help from the medical support. I prefer face-to-face coaching and guidance. Hands-on demonstrations could also help me to learn the position for holding my baby properly for breastfeeding. It is more effective than a Zoom video call on a mobile device … My fear is that I will look back with unhappiness on what I have endured during this time. I think the trauma will be ongoing and far-reaching.”

A typical housing phenomenon in Hong Kong involves many underprivileged people living in tiny “coffin homes” (accommodation in a living unit subdivided into smaller units, each of not more than 150 square feet). Poor living environment, lower educational level, and economic burden impact the emotional stress of participants’ continuation of breastfeeding and affect baby care. Maintaining social distancing among the residents of these “coffin homes” is impossible, and mothers with newborn babies were concerned about becoming infected with COVID-19 easily. Further, due to social distancing restrictions, some participants felt that medical support was becoming remote and only available through online messaging or phone calls.

“My husband is a hawker with leg amputation, and I have a huge financial burden of the family. Contrarily to other families, I have only obtained secondary education level without any professional qualification. Everything made me afraid and anxious, and I do not know how to express. After the completion of my maternity leave, I must return to work as soon as possible. I do not think I would have continued breastfeeding; it was a really big impact for me … I was so stressed because I could not convey and do not know how to convey. Having been seriously disturbed by COVID-19 now, I need to stay in my tiny coffin home with my baby, which is making me worried and anxious, fearing that my baby and I will be infected. I also avoid going to the Maternal and Child Health Center for routine health checks. So, I have not been seeing nurses and receiving their coaching on breastfeeding for a while. I did not realize there was any support elsewhere … My failures in breastfeeding made me feel very frustrated; tears always streamed down my face at night.”

Many participants mentioned that during social distancing restrictions, they could not get together and have physical meetings with other breastfeeding mothers’ support groups. They stated that they wished to connect with others by sharing their experience and seeking support for their breastfeeding. Some participants felt they were alone in the process. Certain participants reported that they were isolated and lacked the emotional support and care from their family members. This was harmful to their physical and psychological health.

“Although my husband was by my side, I still value the presence of my mother accompanying me. My mother was amazing and a strong supporter of breastfeeding. I really hoped she could come to visit me after my son was born. However, in the current situation, she stays in Mainland China and cannot come. Despite being on WeChat [video message], it is not the same as having my mother by my side. I feel like [I and my baby] need her, not just for help, but for emotional support. I also struggle and feel depressed. It seems everything is becoming difficult and frustrating. On the other hand, I want to participate in peer group activities and receive face-to-face support. The lack of support for everything really makes me sad. It is not what I expected. I read a lot of breastfeeding information before my son was born. I could not, however, receive the support I thought would be available.”

However, some participants recognized that staying at home with other toddlers helped reduce some pressure. Others found that although there was no need to take their older children outside, they would still need to help them with online learning (homeschool via Zoom). These daily issues intimidated participants with regard to breastfeeding. Some participants had partners working long hours outside, and who were not able to provide support and help with household chores or caring for the babies. Since the suspension of flights from the United Kingdom to Hong Kong, one participant expressed:

“My mum was not able to fly back and help me with my little son. I now have much less time to focus on breastfeeding. My partner is an A&E (Accident & Emergency) doctor who works from dawn to late night, and sometimes 24 h or more depending on how the workload in the hospital develops. I am at home with an energetic six-year-old boy, who now usually attends classes via Zoom (video message) at home because face-to-face classes are suspended. I do not have time to express my feelings and sometimes I do not even have time to breastfeed my younger son because I feel that I also need to focus and take care of my older son and simultaneously take care of all the household chores. I feel that most of the time, I am like a single mother. In the first three months after the birth of my first son, my mother and sisters returned from the UK to help me at my home. Now, due to the pandemic, I do not have family or friends to help me or to help take care of my elder son so that I can have time to focus on the baby.”

#### 3.2.2. Mothers Being Forced to Focus on Breastfeeding

During the height of the COVID-19 pandemic, compared to other participants who preferred having more or enough time for breastfeeding, several participants felt just the opposite. As they had nowhere to go and not many things to do, they could focus their attention on breastfeeding their babies. Some of them were eager to have a break and do other things to divert their attention.

“My little princess and I were afraid of being infected as if we were in confinement. My focus on breastfeeding is very strong. This, in turn, makes me resent it because it feels like my day revolves around it.”

Breastfeeding is a biological norm; some people still avoid having the taboo of breastfeeding in public, and mothers are still being forced to pump in toilets or storerooms in some shopping malls in Hong Kong. Several participants try to escape complaints or unpleasant experiences of breastfeeding in public; rather, they selected breastfeeding their baby at home.

“I fear men might see. They would image my breasts to seduce … I do not want to breastfeed my baby uncovered in public. Now, during the pandemic, I think it is a good and the best option for me and my baby; I have had a pretty warm and free space for breastfeeding at home.”

#### 3.2.3. Mothers Going Back to “Normal” Life

Different participants had varying views on breastfeeding during the pandemic. Some participants would rather not conduct breastfeeding anywhere except at home. Some participants were concerned that they might miss the unique experience of breastfeeding and lack the opportunity to share the experience and exchange their knowledge of breastfeeding with other new mothers. One of them said that she was anxious and lost in not knowing what will happen in the future, once the COVID-19 pandemic is over.

“If there is a chance, I can breastfeed together with other like-minded mothers in public places, such as the Maternal and Child Health Center, not only in my home. Everyone can share with and coach each other on how to get into the correct position to conduct breastfeeding in various situations and different circumstances. It is a pity that there is now no such get together for happy hours and support. I feel lonely. I do not know how to breastfeed my baby with confidence in public. As such, I might give up breastfeeding earlier than I have planned.”

Some participants resumed work soon after giving birth to their babies. During the pandemic, they were required to work from home. They would have to breastfeed their babies during working hours. In view of the inconvenience and disturbance that this may cause to work, this was an influencing factor for working mothers to give up breastfeeding early. The experience of one participant, who worked in the Accident and Emergency Department in a hospital, was different from those working at home. She stated that she was buried in her work and was required to wear personal protective equipment (PPE) all the time. She was tired, stressed, hot, sweaty, and dehydrated, with no time to rest. This participant could feel her engorged breasts and discomfort and had less milk.

“I work in the A&E unit, and in order to maintain infection control safety measures, I am required to wear PPE all the time. As I do not want to waste the PPE stock, I try to drink as little water as possible to avoid going to the toilet during my shift duty, and hence do not need to change my PPE. I am feeling helpless because of work affecting my breastfeeding planning. I feel like I am pumping less milk at the moment.”

## 4. Discussion

The current study used a phenomenological approach to explore women’s psychological experiences of breastfeeding during the COVID-19 pandemic, specifically in relation to how social distancing measures affected their baby feeding decisions. Two major themes demonstrated how different psychological and difficult experiences emerged in maintaining breastfeeding, alongside the impact that social distancing measures created for them.

The World Health Organization assured mothers that breastfeeding in public is safe, with suitable infection control measures during the COVID-19 pandemic [43]. However, the taboo around breastfeeding in public places in Hong Kong [20] remains. Some participants of this study reflected their worries about breastfeeding newborns while being infected with COVID-19. It is, therefore, important for policymakers to collaborate with healthcare professionals to advance information on social media that helps encourage mothers to continue breastfeeding their babies, despite the worsening COVID-19 situation.

Breastfeeding mothers have been required to face the COVID-19 pandemic and suffer from anxiety [44]; therefore, there was impeded access to healthcare services, with individuals avoiding hospitals and canceling their healthcare appointments [45,46]. The participants’ dialogues in this study, as in some other studies, demonstrated that breastfeeding protects maternal and infant health in both the short- and long-term [47,48,49,50,51,52]. Moreover, breastfeeding protects against infections, providing passive and long-lasting active immunity. Both physical and psychological aspects of mothers are affected by breastfeeding, such as reducing the risk of inflammation, improving sleep distress, and reducing psychological distress [53,54,55]. The findings of this study also indicate concerns regarding their experiences during the COVID-19 pandemic; their mental health is safeguarded when they fulfill their breastfeeding goals [47,48,49,50,51,52,53,54,55]. Conversely, the mothers’ feelings of grief, depression, and trauma increase if they cannot meet their breastfeeding goals.

The findings of this study also demonstrated the mothers’ willingness to discontinue breastfeeding during COVID-19 due to the lack of direct support and reassurance from healthcare professionals, and their relatives. During this critical period, mothers with younger babies are likelier to face varying difficulties and feel uncomfortable with regards to breastfeeding, thus being more likely to discontinue breastfeeding [56,57]. In addition, breastfeeding mothers with older babies expressed that the lack of face-to-face inquiries and support also affected them during the pandemic [58,59]. To avoid contact with other people due to social distancing requirements, breastfeeding mothers expressed that they had negative feelings and stayed at home with limited social contact with people, resulting in declining social opportunities and support services [59]. It is vital to provide professional peer and healthcare support, and communication, during the early weeks of breastfeeding for mothers and their babies, to enhance successful breastfeeding [60,61,62,63]. During the early weeks of breastfeeding, participants of this study who particularly required care terminated breastfeeding due to an absence of practical, emotional, professional peer and healthcare support, and social gatherings. Some participants were also dissatisfied and blamed social distancing requirements during COVID-19, which affected their decision not to continue breastfeeding. Previous studies reported breastfeeding mothers had an increased risk of postnatal depression [64,65,66,67]; our analysis showed few participants planned to extend breastfeeding for a longer time. Importantly, clinicians should track the effects of breastfeeding duration on subsequent physical and mental health outcomes in this population group.

There seems to be a more pressing need for support for those women who are unable to achieve their breastfeeding goals, or who are struggling with a lack of support [68,69]. Studies reported increasing rates of perinatal depression associated with the COVID-19 lockdown in countries worldwide [64,70,71], including worry about health for both mother and baby, professional healthcare support deficits, economic burden effects, strict controls on movement, and social isolation. Policymakers must devise ways to provide holistic support to mothers, babies, and their families whenever they require it [72,73]. According to this study’s findings, the experiences of participants during the COVID-19 pandemic were significantly unique and different. There was an optimistic perception, with most of the participants stating that they had an optimistic impact on their experiences of breastfeeding. They were forced to focus more on breastfeeding their baby during COVID-19 and ensuring social isolation. Participants expressed that they could access enough things for breastfeeding support, such as increasing breastfeeding time for mothers and their babies, had fewer disturbances, kept away from unpleasant opinions, and had extra communication time with supportive partners [65,74,75]. We believe that traditional beliefs and postpartum practices in mothers worldwide mainly involve direct care, dietary intake, and sufficient rest [76,77,78,79]. However, it is very important to pay attention to mothers’ mental health during the current pandemic.

Some studies state that mothers living on low incomes, due to poverty caused by housing factors and lower education levels, are more likely to discontinue breastfeeding in the early weeks [80,81]. Breastfeeding mothers are required to cope with more problems and have less support within the current pandemic. The findings of our study indicated that participants without a bachelor’s degree described having a more undesirable perspective of the lockdown and faced unnecessary hurdles while breastfeeding, especially those who had less support in their living environments. Housing factors, like tiny, sky-high, and dense living spaces, influenced mothers in poorer families who had fewer perceived good parenting qualities, affecting mother-baby relationships [78]. Based on the participants’ experiences, the disadvantages of living circumstances and low educational qualifications both affected successful breastfeeding.

An optimistic impact was observed in participants of this study who were more privileged in their living circumstances, such as having more spacious homes, access to plenty of verdant spaces for exercise and rest, and fewer economic worries. Different components of living circumstances affect the health outcomes of breastfeeding and baby care. Past studies show that mothers having difficulty with the milk ejection reflex showed an association with stress and lactation [82,83,84]. While mothers with poor living circumstances experience higher levels of emotional stress regarding lactation, mothers without living space problems may perceive breastfeeding as a unique and satisfactory experience. Mothers with poor living conditions may experience, for example, the effect of postnatal depression, financial problems, emotional stress, and social isolation during breastfeeding, making the care for their baby more difficult [85,86].

Policymakers, healthcare professionals, and practitioners have been challenged to support a range of creative and resilient responses to strengthen the provision of breastfeeding for both mothers and their babies during COVID-19 [85]. To prevent transmission of the virus, and any other hurdles, behavioral support must be provided for all mothers in terms of positive and accurate information through in-person and telephone services, including telehealth formats through the internet, peer group support, and community resources. Some participants prefer to have more accurate messages through face-to-face breastfeeding support, rather than online delivery support services.

### Limitations and Future Directions

This study is limited as recruitment occurred from support sites; therefore, their experiences do not necessarily represent mothers in general. The research provided strong evidence that social distancing and social isolation, due to Omicron in Spring 2022, have had an adverse cumulative health impact on breastfeeding mothers, both physically and mentally. We used telephonic and Zoom interviews to collect data, which is an increasingly common approach during the COVID-19 pandemic. Nevertheless, our sample excluded participants with significant emotional stress, or a lack of internet connection, which are important factors for discontinuing breastfeeding during the early weeks, according to our findings. While a qualitative study addresses a small group of participants, the findings of our study were not intended to generalize but to explore how mothers understood their perceived reality during the COVID-19 pandemic with regards to practical preparation for breastfeeding, taking care of their child, and supporting their families. Thus, it would be advisable to carry out similar studies in other contexts, such as for different geographic and cultural groups of mothers, to acquire a more thorough understanding of this issue.

## 5. Conclusions

This study used a qualitative approach to explore social support and how COVID-19 impacted mothers breastfeeding their babies during the COVID-19 pandemic in Hong Kong. An analysis of the results showed various positive and negative effects on breastfeeding mothers due to social distancing and social isolation, and different social economic factors, together with their complicated feelings particularly affecting breastfeeding mothers’ feelings of guilt, anxiety, and depression. This occurrence may introduce potential risks to breastfeeding if mothers cannot meet their breastfeeding goals. The results also indicate that policymakers and healthcare professionals should implement tailored strategies to support new breastfeeding mothers, their babies, and families to better prepare them, particularly among socioeconomically disadvantaged groups, to breastfeed in places other than their homes during the pandemic. Further studies should focus on suitable psychological interventions aimed at instilling positive emotions in breastfeeding mothers, alleviating negative emotional states, and protecting their wellbeing in Hong Kong.

## Figures and Tables

**Table 1 ijerph-19-09511-t001:** Participant demographic characteristics.

Characteristic	All	
	n = 20	
	n	%
Age		
≤21	0	0
22–26	3	15
27–31	6	30
32–36	8	40
37–41	2	10
42–46	1	5
≥47	0	0
Education level		
Secondary	2	10
Higher diploma	6	30
Bachelor’s degree	9	45
Master degree	3	15
Number of children		
1	12	60
2	7	35
3	1	5

**Table 2 ijerph-19-09511-t002:** Interview guide.

No.	Probing Questions
1.	How did you feel when you were breastfeeding your baby during the COVID-19 pandemic?
2.	How long have you been breastfeeding your baby? Other than breastfeeding, what other feeding methods have you been practicing for your baby?
3.	Have your feelings changed over time when you were feeding your baby day and night during the COVID-19 pandemic?
4.	Can you tell me how you coped with your own psychological needs concerning breastfeeding?
5.	In what ways do you think your coping strategies have helped you on a psychological level upon breastfeeding experience?
6.	Do you think you are ready to stop breastfeeding, and what makes you think you should stop breastfeeding?
7.	Recalling on your months of breastfeeding, what are your negative and positive feelings towards breastfeeding in your experience during the COVID-19 pandemic?

**Table 3 ijerph-19-09511-t003:** Themes and subthemes of the study.

Themes	Subthemes
Positive influences on breastfeeding support during COVID-19	Mothers’ Desire for Support
	COVID is Not All bad for Mothers
Negative influences on breastfeeding support during COVID-19	Absence of Close and Personal Professional Coaching
	Mothers Being Forced to Focus on Breastfeeding
	Mothers Going Back to ‘Normal’ Life

## Data Availability

The interview guide has been provided in the manuscript using a table. The transcripts that contain private and confidential data, such as the wards and the sites of practice of the participants, will not be publicly available to protect the participants’ privacy.

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
