# Peer review of "The Lived Experiences of Women without COVID-19 in Breastfeeding Their Infants during the Pandemic: A Descriptive Phenomenological Study"

_ijerph, 2022, doi:10.3390/ijerph19159511_

Round 1

Reviewer 1 Report

The focus on this manuscript is on the impact of the COVID-19 pandemic on breastfeeding among non-infected mothers. In general the background is adequately described, however the sentence that ends on line 46 is not complete; also the study described starting on line 68 mentions pre-post comparisons but it is not clear what the pre/post refers to

Aspects of the methods are confusing, in particular how the eligibility criteria (women breastfeeding), 18 month period starting September 2019 and the timing of the interviews that started December 2022. Questions in Table 2 seem awkwardly worded and in some cases not whole sentences (may be it is how they are translated into English?). 

Line 390 - it appears that several worked in the same department which limits the breadth of the sample and potentially compromises the findings   

Line 172 - use described rather than demonstrated

Line 174 - not clear what moral aspect means

Line 195 - 'their' relationship - is that between the parents or between the parent and infant

Unclear when/how member checking occurred

Line 292 - does new mothers mean first time mothers?

Discussion speaks to differences by SES but that is not presented in the results. Conclusions do not mention SES

Line 407 - relevance to the study is not clear

Lines 417-435 - relevance to this study is not clear

Line 453 - do you mean to NOT? continue breastfeeding

Line 455 - not clear why it is policy makers and not clinicians?

Line 471-73 - point not clear

Line 483 - benefits? do you mean support?

Line 485 - sounds judgmental "who have fewer perceived good parenting qualities" 

Discussion includes statements about economics but there are no data/results presented about economics

Line 502 - should it be support rather than 'create'

Line 515 - not clear what "nurture efficiency, educational qualification...." means

Limitations should note that recruitment occurred from support sites so their experience does not necessarily represent mothers in general; the authors point out the lockdown due to omicron in Spring 2022 but do not mention mitigation/lockdown during the period of interest (based on dates provided in the methods section); it is not clear if the mothers interviewed were in lockdowns so again how generalizable is their experience to all mothers or only those living with mitigation procedures. 

In general the manuscript is very wordy. There is a fair amount of extraneous information provided that is not relevant to the rationale or conclusion. 

Author Response

Dear Respected Reviewer,

RE: Manuscript titled "The Lived Experiences of Women Without COVID-19 in Breastfeeding their Infants During the Pandemic: A Descriptive Phenomenological Study"

     Thank you for your unstinting effort to review the revised manuscript. Our research team appreciated your valuable comments. In this document, we provided our responses in a point-by-point format. We hope that our revisions to the manuscript can address your concerns satisfactorily.

     This response letter was prepared with reference to a template that we downloaded from the Journal website. In that template, we noticed that all responses from the authors were highlighted in red. We thus followed the prescribed style and format.

Your faithfully, Corresponding author

Manuscript no.: ijerph-1784525

[The author's name is not shown because a blinded review process is required by the Journal.]

Reviewer 2 Report

This is probably a quite unique paper as there has not been much research into the impacts on breastfeeding of the response to the Covid-19 pandemic. Overall, there is a sound rationale to undertake this piece of qualitative work, and a good sample was achieved and some very lengthy interviews. However, what has been produced lacks the kind of depth and nuance I would expect to see in a phenomenological study. I do wonder whether this is the best epistemological approach for this kind of study given phenomenology’s commitment to describing the essential nature of experience, whereas what is most interesting in the findings is that there were some very contradictory experiences. Would a vanilla thematic analysis not have achieved a similar outcome? Overall, I thin the work needs some significant improvement, and I will try and detail my reasoning for this below.

There is some good background in the introduction, although it could be more concise. In order to understand some of the findings more effectively and to understand the context better, it would be useful to include more details about the nature and context of breastfeeding in Hong Kong. For example, what are the forms of support available, what are the rates of breastfeeding, average length of sustained breastfeeding, inequalities in breastfeeding rates, etc? What are general public attitudes towards breastfeeding and in particular breastfeeding in public spaces (as this gets mentioned by participants)?

What language used for interviews and, if not English, how was translation managed?

The interview questions appear either poorly translated or lacking development that would enable participants to provide in-depth accounts of their experiences. They seem quite disjointed and do not suggest that they would create a ‘conversation with a purpose’ approach that if often the aim in qualitative interviews.

The analysis process described is ok, although it is not very clear how the authors moved form this to the themes presented.

Table 3 is suggestive of what Barbour (2001) warned of as a tick box approach to ensuring ‘rigour’ in qualitative research. It would be better to have this as an appendix, and to give brief examples of what was done, e.g. how were field notes used to analyse transcripts? The use of inter-coder reliability is suggestive of some kind of underpinning fundamental truth that is being reached by coding, which contradicts the epistemological assumptions of qualitative research (Braun & Clarke  2022 have discussed this in some depth in their most recent book).

Some of the findings are confusing and need reconsideration. The two themes seem very obvious, lack depth, and are not very interesting, but the short excerpts suggest there is much more going on in the data. There are clearly interesting contradictions between some who were very happy with the lack of social contact as they felt this gave them space to breastfeed in peace without disturbance or interference from family and friends and others who (very specifically) felt that the absence of the advice and guidance from their own mothers was hindering their ability to breastfeed effectively. There are also indications that there are some very clear inequalities here, which is touched upon in the discussion. Is there not more in the data about this? Very little on the wider psychological impacts on the mothers of lockdown and how this impacted on their breastfeeding. There are interesting findings on the restricted access to social/professional support and the potential for this to result in early cessation of breastfeeding.

Overall, I wanted more from the findings and was quite frustrated by their quite descriptive and thin nature.

The discussion starts with Covid-019 mother to baby transmission, but this is not something considered in the data these were non-infected mothers. Was this something that the mothers were concerned about, as it does not come through very clearly in the findings?

Much of the discussion makes quite large claims on the basis of some relatively thin evidence. It may be that there is some good data in the interviews, but it has not been brought out very well in the findings.

Overall, I feel that this needs some significant revision before it would be suitable for publication.

References

Barbour R. S. (2001). Checklists for improving rigour in qualitative research: a case of the tail wagging the dog?. BMJ (Clinical research ed.)322(7294), 1115–1117.

Braun, V. and V. Clarke (2022) Thematic Analysis: A Practical Guide. London: Sage.

Author Response

Dear Respected Reviewer,

RE: Manuscript titled "The Lived Experiences of Women Without COVID-19 in Breastfeeding their Infants During the Pandemic: A Descriptive Phenomenological Study"

     Thank you for your unstinting effort to review the revised manuscript. Our research team appreciated your valuable comments. In this attached document, we provided our responses in a point-by-point format. We hope that our revisions to the manuscript can address your concerns satisfactorily.

     This response letter was prepared with reference to a template that we downloaded from the Journal website. In that template, we noticed that all responses from the authors were highlighted in red. We thus followed the prescribed style and format.

Your faithfully,

Manuscript no.: ijerph-1784525

[The author's name is not shown because a blinded review process is required by the Journal.]
